# Trapping Brown Marmorated Stink Bugs: “The Nazgȗl” Lure and Kill Nets

**DOI:** 10.3390/insects10120433

**Published:** 2019-11-30

**Authors:** David Maxwell Suckling, Valerio Mazzoni, Gerardo Roselli, Mary Claire Levy, Claudio Ioriatti, Lloyd Damien Stringer, Valeria Zeni, Marco Deromedi, Gianfranco Anfora

**Affiliations:** 1Technology Transfer Centre, Fondazione Edmund Mach, I-38010 San Michele all’Adige, Italy; gerardoroselli@hotmail.it (G.R.); Claudio.Ioriatti@fmach.it (C.I.); 2The New Zealand Institute for Plant and Food Research Ltd., PB 4704, Christchurch 8140, New Zealand; Lloyd.Stringer@plantandfood.co.nz; 3School of Biological Sciences, University of Auckland, Auckland 1072, New Zealand; 4Research and Innovation Centre, Fondazione Edmund Mach, I-38010 San Michele all’Adige, Italy; valerio.mazzoni@fmach.it (V.M.); valeriazeni93@gmail.com (V.Z.); Marco.Deromedi@fmach.it (M.D.); Gianfranco.Anfora@fmach.it (G.A.); 5Biological and Biotechnical Control Agency, 00123 Rome, Italy; 6Center of Agriculture, Food and Environment (C3A), University of Trento, I-38010 San Michele all’Adige, Italy; 7Kallisto, Mt Pleasant, Christchurch 8081, New Zealand; ClaireLevy@xtra.co.nz

**Keywords:** aggregation pheromone, *Halyomorpha halys*, trap, ghost trap, hedgerow, lure and kill

## Abstract

Improvements to current brown marmorated stink bug (BMSB), *Halyomorpha halys*, surveillance and killing systems are needed to improve detection sensitivity and to reduce pesticide use. Detection of BMSB in New Zealand with traps is reliant on sticky panels with aggregation pheromone, which are low cost but inefficient compared with beating foliage. Trapping for BMSB adults and nymphs was conducted daily with lethal traps consisting of an aggregation pheromone-baited-coat hanger covered with dark-colored long-lasting insecticide-treated mesh, we termed “The Nazgȗl”, based on its sinister appearance. A deep tray lined with white plastic was attached centrally at the base for collecting the dead BMSB. The lethal traps killed and caught up to 3.5-fold more nymphs and adult BMSB than identically-baited sticky panels in the 3 weeks of deployment, and provided a snapshot of phenology by instar. We expect that lure-and-kill stations could contribute to the suppression of a delimited population and could be included as part of a semiochemical-based eradication program. Attracting and killing females and nymphs, thus removing future offspring, could contribute to population suppression during an eradication.

## 1. Introduction

*Halyomorpha halys* (Stål, 1855) (Hemiptera Pentatomidae), the brown marmorated stink bug (BMSB), is an invasive and highly polyphagous species from Southeast Asia. With its hitch-hiking behavior and rapid growth rate [1], swift recent expansion in its geographic range into the United States of America, Europe, and South America has been observed. BMSB feeds on a wide range of plants, including field crops, vegetables, trees, fruit, and ornamentals. Affected plant tissues react, and the feeding punctures usually lead to the formation of necrotic areas and deliquescent fruit pulp [2]. Pre-reproductive adults over-winter and are a nuisance inside human houses because of their abundance and unpleasant odor [3]. In northern Italy [1], BMSB is predicted to complete two generations per year [4] as in the United States of America [5].

The BMSB is increasingly widespread in Europe [2,6]. The first record of BMSB in Italy was near Genoa in 2007, and the first detection of an established population was in Modena Province in September 2012 [7]. Interceptions of this species have been reported in other countries [8], including New Zealand [9]. Adult BMSB is estimated to have the capacity to fly 2–3 km per day [10] and generally has a high natural dispersal ability [11,12,13], which is likely human-assisted considering how fast it spread across Europe and North America. Governments considering eradication of arthropods are facing increasing challenges from novel pest biodiversity [14]. Avoiding the long term problem of BMSB by eradication warrants investigation where this might be possible [15], but this approach requires the development of suitable tools [16]. No eradication attempts have been reported thus far, but New Zealand has taken the innovative and proactive step of approving an exotic biological control agent (*Trissolcus japonicus*) for release to support an eradication attempt of BMSB [17]. Host range testing was conducted on several non-target pentatomid bugs [17]. New Zealand has an official government goal of harnessing 4.7 million pairs of eyes for biosecurity [18], and has a long and successful track record of unwanted arthropod eradications [19], as well as a desire to avoid non-target impacts from new parasitoids where possible [20], leading to our search for possible alternative eradication technologies, such as the sterile insect technique [21].

Aggregation pheromones have been identified [(3*S*,6*S*,7*R*,10*S*)-10,11-epoxy-1-bisabolen-3-ol and (3*R*,6*S*,7*R*,10*S*)-10,11-epoxy-1-bisabolen-3-ol] [22] and subsequently improved with the addition of methyl (*E*,*E*,*Z*)-2,4,6-decatrienoate (MDT) [23]. Various trapping systems have been investigated for Integrated Pest Management (IPM) [24], and based on these lures, surveillance systems have been tested in the context of border protection [9]. The lures have also been tested for lure and kill at high doses (84 and 840 mg) in sacrificial insecticide-treated trees [25]. Tree-level attract and kill has the potential to reduce damage in orchards, but so far it is very expensive [26]. We chose to use sticky panels as a reference system due to extensive previous testing [27], for comparison with alternative traps under an expanding BMSB population in the north of Italy. While there is no proposal to use sticky traps as a control tool using mass trapping, our goal was to determine what improvements could be made to support an incursion response. Novel systems are especially needed for border protection in countries with high interception rates, such as New Zealand [9], where interceptions of alien pests are often detected in urban or peri-urban areas.

From a pest management perspective, knowing how the pest population moves into a crop is fundamental because it could help farmers to decide when and where to apply insecticides or other tactics to prevent economic losses [28]. Insecticide net systems have been developed for use outdoors in forestry [29], and the use of long-lasting insecticide nets has been under investigation for protecting crops from immigration by BMSB, although there have been issues, including poor efficacy at preventing damage [30,31,32]. The netting has also been investigated for reducing BMSB as a nuisance pest [3].

Free-standing white “ghost traps”, which employ the aggregation pheromone and long-lasting insecticide nets, have been tested for pest management [32,33,34,35]. This type of lure-and-kill approach has benefits of limiting broadcast insecticide usage, thus is compatible with biological control, but has ongoing labor costs [36]. For jurisdictions conducting surveillance, such as New Zealand, new and effective methods for both surveillance and suppression are important. The goal was to identify which sampling technique using the pheromone had the greatest sensitivity for surveillance, and to investigate the potential for ghost traps [33,34]. To remove the need for structural support of the net and to move the device closer to probable bug sources, we suspended the pheromone-baited insecticide-treated nets in non-crop trees, differentiated here as “The Nazgȗl” (Figure 1). Although we have recently investigated peri-urban social attitudes to the sterile insect technique [37], we have not done so with the new traps. While we have not investigated community responses to The Nazgȗl, we imagined that J.R.R. Tolkein’s terminology [38] might be acceptable to the New Zealand public at least, who are sympathetic to protecting their country, which is essential for biosecurity.

## 2. Materials and Methods

The first experiment aimed to estimate the catch efficiency of BMSB from the two trapping and pheromone devices most frequently used commercially, Pherocon sticky panels and Rescue!^®^ traps baited with standard or high dose lures (below). The second experiment was designed to evaluate these traps against beating tray efficiency. In Experiment 3, we used transects deploying The Nazgȗl in an attempt to reduce hedgerow populations of BMSB, assessing populations using beating tray samples to avoid population interference. Finally, in Experiment 4, the kill efficacy of Nazgȗl was compared with sticky panels, using the same lures (normally used for monitoring), and post-insecticide exposure recovery was investigated.

### 2.1. Experiment 1. Field Test of Traps and Lures with Adults at a Forest Margin

This experiment aimed to estimate the catch efficiency of BMSB adults and nymphs from the two trapping devices most frequently used commercially. Trapping was conducted in August and September 2018 at Fondazione Edmund Mach (46°11′46″ N, 11°8′11″ E), San Michele all’Adige, TN, Italy. Trap catches were evaluated with two different kinds of traps; Rescue!^®^ Stink Bug Trap (Sterling International Spokane, WA, USA) and Pherocon sticky panels (Trécé, Adair, OK, USA), in combination with two loadings of pheromone. Standard and high dose lures (Trécé, Adair, OK, USA) contained either 5 mg and 50 mg or 20 mg and 200 mg (i.e., 4 × loading) of the two pheromone components: (1) (3*S*,6*S*,7*R*,10*S*)-10,11-epoxy-1-bisabolen-3-ol and (3*R*,6*S*,7*R*,10*S*)-10,11-epoxy-1-bisabolen-3-ol [22] plus (2) methyl (E,E,Z)-2,4,6-decatrienoate that works synergistically with the bisabolenes to attract BMSB to traps [23,27].

We set four different combinations of trap (Rescue trap or sticky trap) and lure (Trécé Standard or high load lure) (n = 5 replicates) at a forest margin with adjacent vineyards of mixed grape varieties. Lures were positioned on top of the sticky panels or inside the Rescue traps. In addition, there was also a ~10 cm × 5 cm piece of mesh permeated with alpha-cypermethrin (1.57 mg ai/g fiber, Storanet^®^, BASF, Ludwigshafen, Germany) placed inside the Rescue traps. The experiment was set up on 7 August and checked approximately every 10–11 days for 40 days (four checks). The denominator degrees of freedom (calculated by subtracting the number of sample groups from the total number of samples tested) was 16.

### 2.2. Experiment 2. Field Test of Catch Rates in an Orchard Hedgerow

This experiment was designed to compare trap efficiency with beating tray efficiency. One 200 m long hedgerow at the Fondazione Edmund Mach research farm at Rovereto, TN, Italy (45.8780 N, 11.0197 E) was divided into 20 plots (separation ~10 m). Each plot contained a trap located at a convenient height of 1.8 m above ground level. Traps alternated between Rescue and sticky panels with high loading lures. The hedgerow consisted of mixed deciduous trees dominated by hazelnuts (*Corylus avellana* L.) and plums (*Prunus avium* L.) with *Crataegus monogyna* Jacq*., Arbutus unedo* L., *Populus* sp., *Clematis* sp., and *Euonymus europaeus* L. Vineyards were present on both sides of the hedgerow for several hundred meters. The 20 plots in the hedgerow were pre-baited with standard-dose pheromone lures for 1 week to boost background numbers of BMSB. These lures were removed 24 h before the trial commenced.

Sticky panels and Rescue traps were established from 22–27 August 2018 (Run 1) and from 12–17 September 2018 (Run 2). Trap catches were recorded after 0, 2, and 5 days. In addition, two cloth beating trays (metal frame dimensions 40 × 80 cm) were operated continuously for 1 minute per plot (0.5–5 m either side of the lure) at 0, 2, and 5 days. Catches were pooled from the two beating trays to give a single value per plot because of some variation in team personnel sampling efficiency, which we did not quantify. 

### 2.3. Experiment 3. Field Test of The Nazgȗl in Orchard Hedgerows

We cut alpha-cypermethrin-impregnated netting as mentioned above into 1.5 × 2 m rectangles and baited each piece at the top with high dose lures for BMSB (4× dual lures, Trécé, Adair, OK, USA). The manufacturer’s claimed outdoor insecticidal longevity of the netting is two years, and our trials were all short duration. The netting and lures were draped over and tied to inexpensive wire coat hangers (Nano non-slip hanger, Eurobazar, SRL, Arco, Italy), to increase the surface area exposed for landing and walking from nearby foliage. The Nazgȗl were suspended from hedgerow tree branches at about 1.8 m above ground level, chosen for convenience. The three hedgerow sites were surrounded by orchards in the Trentino Region, Italy, and were under 4 m in height. The bases of the nets were tied with string to the inner centers of rectangular plastic food boxes (c. 40 × 50 cm) lined with white plastic and designed to catch falling bugs. The hedgerow at Site 1, at the Fondazione Edmund Mach research farm at Rovereto described above, had been reduced to ~80 m length in late 2018, so an untreated control hedgerow was chosen nearby, dominated by plums and fruiting elderberry (*Sambucus nigra* L.). Site 2 was a continuous hedgerow at a private organic apple orchard dominated by fruiting elderberry and *Cornus mas* L., and Site 3 was at a private conventional apple orchard, dominated by a continuous row of fruiting *Viburnum lantana* L. and elderberry. The Nazgȗl traps were accompanied by plastic-covered signage, announcing “Non Toccare” (Italian for “don’t touch”) with a brief explanation of the insecticide netting and the experimenter’s contact details. 

Beating tray samples were conducted as per Experiment 2, with the BMSB counted and adults identified to sex and nymphs to life stage before being released again. This was performed at each of the three hedgerow locations, with 10, 12, and 18 beating samples taken at 10 m intervals at sites 1–3, respectively, immediately before the establishment of The Nazgȗl traps on 22 July 2019 in one half of each sampled area (split-plot design). After the establishment of hedgerow transects with 5, 6, or 9 of The Nazgȗl traps at 10 m spacing (the number depended on hedgerow transect length available), counts of dead or dying BMSB on the netting or in the trays were made daily for four days and again after nine and ten days. The insects were removed at each sampling event (24 h after emptying the trays in each case; traps were cleared at Day 8 to achieve this on Day 9). Beating was repeated after two and nine days, and the traps removed. Nymphs were counted separately by instar, and adults by sex for both the beating tray, as well as for The Nazgȗl. Lures were not added to the control plots, to avoid increasing the probability of economic fruit damage for the grower. The trial was removed after it was apparent that the kill rate was less than the attraction rate.

### 2.4. Experiment 4. Comparison of The Nazgȗl Catch Rates with Sticky Panels at the Forest Margin and Recovery from Exposure to Insecticide

A field comparison of sticky panels (Experiment 1) and The Nazgȗl (Experiment 3) was conducted at the above-mentioned margin of the forest and vineyards at Fondazione Edmund Mach, San Michele all’Adige. The two trap types were alternated, and each trap was separated by 10 m (nine replicates of each). Traps were checked daily from 5–26 August 2019. The same lures as above were used throughout (4× dual lures, Trécé, Adair, OK, USA). Nymphs were counted separately by instar and adults by sex for both the sticky panels and The Nazgȗl. Adults from The Nazgȗl were caged with food and water under natural light at 24–26 °C on 17 days, and the mortality assessed daily for the next six days to estimate recovery after sub-lethal exposure in the potentially reversible toxicological phenomenon called “pyrethroid knock-down” [39]. Untreated control BMSB were captured daily elsewhere in the forest margin and caged similarly (untreated: n = 13–17 BMSB per day, Nazgȗl: n = 21–42 BMSB per day).

### 2.5. Statistical Methods

Experiment 1. The factors were trap type (Rescue or sticky), pheromone load (standard or high) and date of sampling (n = 4; 7 and 17 August; 10 and 21 September) and the response variable was the number of BMSB caught in traps (“trap catches”). Correspondence analysis (CA) [40] was carried out using the factors as column variables, and capture numbers as the row variable. Three categories of capture were defined: zero (no captures), low (≤8 captures) and high (>8 captures). To distinguish between low and high captures, the median value of 8 captures was chosen as the threshold. Sites (n = 5) were used as supplementary column points. A full factorial linear mixed model was performed on the capture data, using trap type and pheromone load as fixed factors and date of sampling as the random factor. Capture data were log-transformed to meet the assumptions of normality and homogeneity.

Experiment 2. A full factor analysis of variance (ANOVA) was carried out on the factor’s trap type (sticky or Rescue), method of capture (trap or beating), and period of sampling (run 1 or run 2) for the response variable catch, which was log-transformed to meet the assumptions of normality (and homogeneity). Boxplots show the median and quartiles with whiskers for adjacent values that are the lowest and highest observations that are still inside the region defined by the following limits: lower Limit: Q1 − 1.5 (Q3−Q1), Upper Limit: Q3 + 1.5 (Q3−Q1).

Experiments 3 and 4. For the two Nazgȗl trials (Experiments 3 and 4), counts of nymphs were converted to adult equivalents by multiplying the count of each stage by survivorship estimates [41] (N5: 0.97, N4: 0.92, N4: 0.75, N2: 0.65, N1: 0.61), and the sum calculated across nymph stages after the adjustment for probability of reaching adulthood for each stage. These values were log-transformed to generate an approximately normal distribution to stabilize the variance. For the general linear model of beating tray counts in Experiment 3, adjusted counts for each sample location were analyzed for site (farm), treatment, and date (before and after treatment at 2 and 9 days). Adjusted adult counts were also used for the ANOVA of effects from alternating trap type on the catch in two runs of Nazgȗl and sticky panels. The percentage change in mortality was investigated for the trial on recovery from exposure to the insecticide, with the data corrected for control mortality using the Schneider–Orelli formula (% mortality in Treated—% mortality in Control/(1—% mortality in Control) [42].

## 3. Results and Comments

### 3.1. Experiment 1. Field Test of Traps and Lures with Adults

Traps were successful at sampling the incipient BMSB population at San Michele in 2018, but catches were affected by pheromone loading and trap type (Table 1 and Table 2).

Pheromone load (Ph) was the most significant factor, followed by the trap-interaction (Table 2).

### 3.2. Experiment 2. Field Test of Catch Rates in Orchard Hedgerows

The ANOVA on log-transformed values was highly significant for all three factors (Appendix A). The beating tray samples from the hedgerow of hazelnut trees consistently produced high numbers of adult BMSB. Counts for the two runs, taken 2 weeks apart, were significantly different from each other (*p* < 0.001), so they have been presented separately (Figure 2). This difference was mainly because of the better relative performance of the sticky panel in the second Run (*p* < 0.01). There was a higher number of catches by beating in comparison to the sticky panels, in Run 2 (Figure 2). Catches in sticky panels were significantly higher than the Rescue traps, and this effect was greater in the second run due to the improved performance of the sticky panels (Figure 2). On all occasions, beating was significantly more effective at collecting BMSB. Beating highlighted a greater catch around the sticky panels where the lure was placed in the open air, unlike near the Rescue traps that contained the lure, potentially impeding the release rate.

### 3.3. Experiment 3. Field Test of The Nazgȗl in Orchard Hedgerows

Beating tray sampling, completed before the installation of The Nazgȗl in the treatment plots at the three isolated hedgerow transects surrounded by orchards and vineyards, produced similar results to samples taken in the control plots (F_1,4_ = 0.12, *p* = 0.744, Figure 3). However, the installation of the high dose lures increased the counts of adult BMSB, as shown by the beating counts 2 days later at Sites 1 and 3 (F_1,4_ = 4.58, *p* = 0.099, Figure 4). Sampling time at 2 and 9 days post-treatment showed highly significant increases in the adjusted adult counts, and treatment was also highly significant, with a lower level of significance for site 2 (Figure 4, Appendix A). The sex ratio of BMSB adults trapped over this period was about two females to one male in both The Nazgȗl (211 males: 418 females) and beating trays (157 males, 308 females).

The median adult bug counts from beating trays were 7-fold higher than in the untreated plots after two days and 6.25-fold higher after nine days of The Nazgȗl present (Figure 4). The highly mobile adults showed the greatest increase in counts after the aggregation pheromone-baited Nazgȗl were installed; numbers of nymphs caught by beating remained low and similar in both treatments and controls (lower). Over the six readings (taken over ten days), The Nazgȗl in the hedgerows produced mean kills of 24.7 males and 56.8 females per trap, as well as 1.9 5th instars, 2.6 4th instars, 3.1 3rd instars, 25.8 2nd instars, and 5.2 1st instars. The effective daily kill rate increased to a mean of ~25 adult equivalent bugs per day, by day 10 (Figure 5, or about the equivalent of an egg batch per day), under the BMSB density in the area.

### 3.4. Experiment 4. Comparison of the Catch with The Nazgȗl and Sticky Panels

A two-way ANOVA showed no significant effect for date, so a one-way ANOVA was conducted on log-transformed summed catches across all dates, which were significantly greater for The Nazgȗl (F_1,16_ = 10.37, *p* < 0.005). Catches in sticky panels remained low and constant while there was an upward trend for the number killed attributed to The Nazgȗl (Figure 6).

At the forest edge, The Nazgȗl had reported about 3.5-fold more nymphs than detected on the sticky panels, and consequently, there was much more information on population changes by life stage from The Nazgȗl trap, especially for 2nd instars and females (Figure 6), which highlights the differences with sticky bases, which in this trial did not reflect changes in density.

All mobile life stages were converted to adjusted adults for both trap types, and there was a 2.6-fold higher catch of adjusted adults in The Nazgȗl (up to a mean of 15 BMSB killed per day, Figure 7, or about half the rate seen in Experiment 3). Data for the prototype Nazgȗl (Figure 1) indicated a mean of 52 BMSB killed per day over 52 days (see Graphical Abstract). After the first 3 days, there was no strong surge in the catch over time in this experiment, but an upward trend remained. The numbers of adjusted adults did not reach as high as at the hedgerow and apple orchard sites further south. Catches of adult BMSB in The Nazgȗl or sticky panels were the same in the first 24 h, but the standard errors in the catch of adults and nymphs did not overlap between The Nazgȗl and the sticky panels on the majority of dates, thereafter. The ANOVA for treatment was significant (Appendix A).

The mortality of adult BMSB removed from The Nazgȗl in the field to the laboratory was consistent over six days, after a slight reduction (−15%) in mean mortality for the BMSB previously-exposed to insecticide in The Nazgȗl trap (Figure 8), due to recovery from knock-down (Figure 8). There was a mean increase in mortality of untreated control BMSB in 24 h (18%). Taken together with the mortality seen in the untreated control, this suggests a slight bias towards survival and recovery for a low proportion of the poisoned BMSB, up to about 25% after 24 h (Schneider–Orelli formula).

## 4. Discussion

Efficient traps can support management decisions to restrict the use of insecticides, which can lead to a reduction in costs, non-target effects, and secondary pest outbreaks [43]. Invasive insect species can severely disrupt established IPM programs [44], and monitoring new pests can be difficult due to a general lack of knowledge about their behavior and ecology. One approach to avoid the long-term costs of pests is to undertake the process of eradication, which has a surprisingly high success rate against some pest groups [45]. While expensive and rarely possible, this can be the least cost option if there are sufficient tools available for delimitation and suppression to near extinction, allowing Allee effects to take over at low density [15]. However, the characteristics of suppression tools that would be effective in an eradication, potentially in an urban area, are likely to show some differences from pest management applications in orchards [37,46], so any alternative to broad-cast insecticide applications warrants consideration.

Unlike the ghost traps, that are shown as free-standing in photographs [33], The Nazgȗl were suspended directly from branches with a coat hanger and could be moved higher if needed. If they were to be deployed in urban areas, The Nazgȗl might need to be suspended out of reach of the public. The presence of the pheromone and the folds of treated netting was an effective combination in the open field, unlike insecticide panels, which were previously found ineffective without the pheromone [3]. Limited exposure to the insecticide was attributed to the poor kill rate in previous studies, and the present study found ~25% recovery from knock-down when the correction for control mortality was applied, acknowledging that bugs were removed back to the laboratory which could have shortened their exposure. Exposure and response would likely depend on arrival time, duration of exposure to the netting, and temperature. The survivors apparently tended to be larger adults, with a bias towards females, which are frequently larger, although this was not quantified. There is an inverse relationship between median lethal dose and temperature in pyrethroids [47] so that toxicity for bugs on the mesh at night would likely be greater under cooler temperatures. This also means that mortality would vary seasonally, and could be greater in spring and autumn than in mid-summer. Improvements to the netting, such as making its microscale texture hairy, might help to increase mortality. Further changes in mortality after 1 day were not seen from days 2–6 in the insects recovered from The Nazgȗl, although control mortality rose relatively steadily during that time. Eggs were laid in cages housing the controls, and some oviposition was also seen in the BMSB recovered from The Nazgȗl, although these females may have been only lightly exposed given the timing of overlapping generations of adults in the field during the trial in August, this could have resulted from the greater background mortality of older overwintered adults as well as their greater of susceptibility to insecticides, such that the younger individuals were expressing recovery in the treatment but the older individuals, seen dying daily in the controls, were potentially already killed in The Nazgȗl traps. 

The first trapping experiment was limited to a single hedgerow, and this lack of replication at the hedgerow scale means that the data could not be extrapolated beyond our replicated sampling unit [48]. The hedgerow study did show that lures increased the number of BMSB seen by beating, especially to sticky panels where the lures were exposed (unsurprising). The study was useful to confirm the most efficient combination of lure and trap under local conditions, and the results unsurprisingly confirmed an improvement from the higher loading, reported by Acebes-Doria et al. [49]. The results helped with the development of the first hedgerow transect study, which showed significantly greater sensitivity of beating trays over traps while avoiding the potential bias caused by the attraction of BMSB to the pheromone. The trap comparison aimed to understand which combination was the most efficient for IPM monitoring programs and to evaluate the population in the hedgerow transect study using sticky panels, which have been well studied elsewhere. The low efficiency of both trap types likely obscured some differences, although the Rescue trap caught more BMSB if there was a very high surrounding population. Sticky panels operated season-long in the United States of America peaked at up to 10–20 for all stages of nymphs per week in the most abundant locations [50], which is much lower than seen here with The Nazgȗl (see graphical abstract also, which peaked at 100 bugs killed per day). Of course, sticky panels are not normally considered for suppression.

It was not possible to separate the effects of presenting significant amounts of pheromone (20 stations placed every 10 m in a 200 m transect) from the odors from ripening fruit of preferred host plants (e.g., hazelnuts) or other effects inducing the BMSB to remain in or colonize the hedgerows. The trials were sufficiently short that these effects probably did not change much. It is important to know the extent of dispersal of BMSB on the landscape, and the key drivers affecting it, to create a proper pest management program. To reduce BMSB populations, it may be possible to use the pheromone and knowledge of the attractive hosts in an area, to concentrate the population for intervention outside the crop, but the fast movement of the insects between hosts means that such a tactic would need to be integrated with local knowledge of host phenology. The plume reach of the pheromone has been estimated as 40 m [28], but this was not studied here. A forthcoming study has expanded knowledge of the impact of fruit volatiles on dispersal and host location [50], which clearly play a role [51].

In Experiment 3, the desired result of a reduction in numbers of BMSB was not seen in the three treated hedgerows, in fact there was a significant increase in BMSB sampled by beating trays in the treated areas (Figure 3 and Figure 4), despite a daily removal of adult and nymphal BMSB (Figure 5). The experiment was, therefore, discontinued when it became apparent that the attraction of BMSB to the baited hedgerows was potentially greater than the number of BMSB being killed since this could risk increasing fruit damage on the adjacent orchards. These results suggest that locally baiting such hedgerows could potentially worsen pest damage locally unless deployed over a much larger area. However, it was not possible to determine whether concentrating the BMSB in hedgerows was really having a net negative effect, or whether the net effect of concentrating and killing the BMSB from the nearby crop was advantageous to the growers by reducing bugs elsewhere. A much larger-scale trial of longer duration would be needed for this, similar to trials in the United States of America [26]. Local removal of a portion of the next generation of nymphs would appear to be a benefit, and removal of the order of an egg batch a day per trap appeared to be happening after the start-up period. Reports from the United States of America suggest that long-lasting insecticide nets have generally not reduced fruit damage (G. Krawczyk pers. comm.), but improved deployment systems with high treatment density might warrant investigation for delimited populations.

At the forest–vineyard margin, the numbers of BMSB attracted and killed were higher for The Nazgȗl than for the sticky traps. Observation revealed the tendency of walking nymphs to detect and avoid the sticky glue. In contrast, nymphs formed a high proportion of individuals killed by The Nazgȗl. The larger surface area of the netting over the sticky panel would also play a role in the catch. Further, the netting was in contact with multiple tree branches allowing easy access for nymphs. The adjustment of nymph counts by the probability of reaching the adult stage enabled the counts to be readily compared between treatments as a single number. The Nazgȗl trapping counts potentially suffered some unseen losses in counts from rain or wind affecting the narcotized or dead bugs falling outside the trays, although the tray sides appeared to be high enough (15 cm) to minimize this effect most of the time, noting that dead individuals were visible nearby traps with an absence of vegetation. Losses (killed but not counted) were not assessed, and improvements to reduce losses would require further investigation. We also observed adult BMSB that were on or adjacent to The Nazgȗl, which escaped sampling by flying away on approach from the field team.

In both Experiments 3 and 4, the number of bugs arriving and being killed increased over the first three or so days, although the levels were different in the orchard hedgerows and forest margin, suggesting higher background populations in the hedgerows. For logistical reasons, the study at the forest margin (Experiment 4) was able to be conducted for longer with daily readings, and the lack of drop-in catch over time to The Nazgȗl suggests that the removal of BMSB did not depress the population. In fact, there was a build-up of second instar nymphs over time, possibly because of the attraction of females and localized oviposition near the traps. The Nazgȗl appear to provide useful resolution of the phenology, without apparent bias for life stages. In contrast, the catches to sticky bases were independent of the catches by The Nazgȗl, which were 1.4–10.4 higher. Looking at false negatives, there were 12 cases of zero bug catch in 24 h with the sticky panels and 3 cases of zero bug catch in 24 h with The Nazgȗl, over the three weeks. The Nazgȗl have not been compared directly with a novel live trap, capable of catching seven-fold or higher numbers of live adult BMSB than sticky panels [52]. Live traps present the possibility of the wild harvest of BMSB for the sterile insect technique [53].

## 5. Conclusions

Such killing stations, as demonstrated by The Nazgȗl, could prove useful in suppression or eradication of a delimited population, with or without the trays to determine the numbers killed. Sticky panels with high dose lures are currently preferred for cost-effective surveillance, and proactively-established killing stations could complement this in high risk sites. While the present study of the potential of The Nazgȗl for killing adult and nymphal stages of BMSB is relatively preliminary (and we await further publication of results especially from IPM researchers in the United States of America studying factors such as net longevity [35]), it can be concluded that this lure-and-kill concept could be compatible with the needs of an eradication to reduce numbers of a known and delimited population, while avoiding broadcast insecticide use, where this is a sensitive issue. Proactive deployment of suitable numbers of The Nazgȗl at high-risk sites, such as ports or devanning locations, could prove useful to reduce the risk of establishment. It has been further reported that higher loadings of pheromone from the inclusion of multiple lures can increase the efficacy of ghost traps (G Krawcyzk pers. comm.), which is consistent with work on attract and kill using apple trees [26]. The limitations of the active space of the aggregation pheromone and the cost of the materials will require careful consideration if considered for use in eradication, but warrant further exploration given the paucity of tools for this daunting task, which has yet to be attempted by any government against this insidious pest. The pro-active registration of this type of long-lasting insecticide net material is advised, as it represents a novel product in some jurisdictions.

## Figures and Tables

**Figure 1 insects-10-00433-f001:**
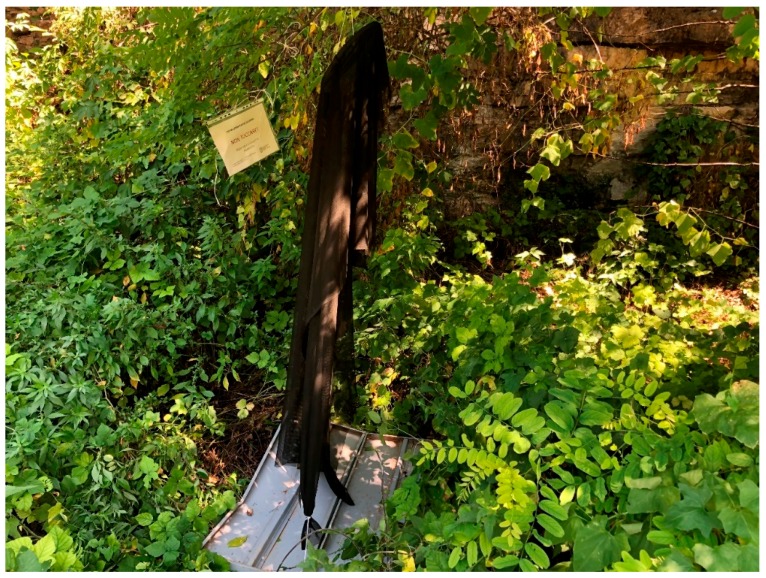
Prototype Nazgȗl trap at the forest–vineyard margin constructed at the Fondazione Edmund Mach, San Michele all’Adige, (TN), Italy. A coat hanger was used to suspend the aggregation pheromone and the 1.5 × 2 m Storanet long-lasting insecticide net that was fixed above the insect catch tray. In subsequent trials, the catch tray was replaced with white plastic-lined boxes. Catch data from this prototype are presented on the graphical abstract (total kill 1983 BMSB 5 July–26 August 2019, 52 days).

**Figure 2 insects-10-00433-f002:**
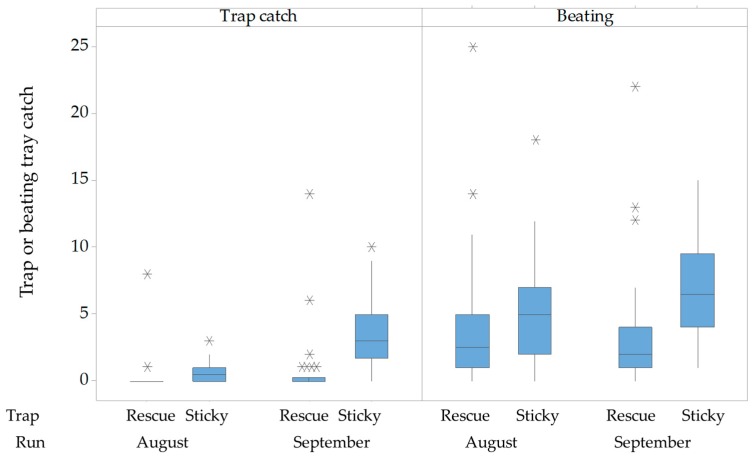
Boxplot of *Halyomorpha halys* catch per date from Rescue and sticky traps (left) paired on the right with catches from adjacent (10 m apart) beating trays, in two runs in a hazelnut hedgerow (n = 10 replicates for each trap in each run, August and September) from Experiment 2. Inner lines are medians. The box includes 25–75% of the data, and stars are outliers beyond this. Whiskers are defined in the Methods. Traps were baited with high dose lures.

**Figure 3 insects-10-00433-f003:**
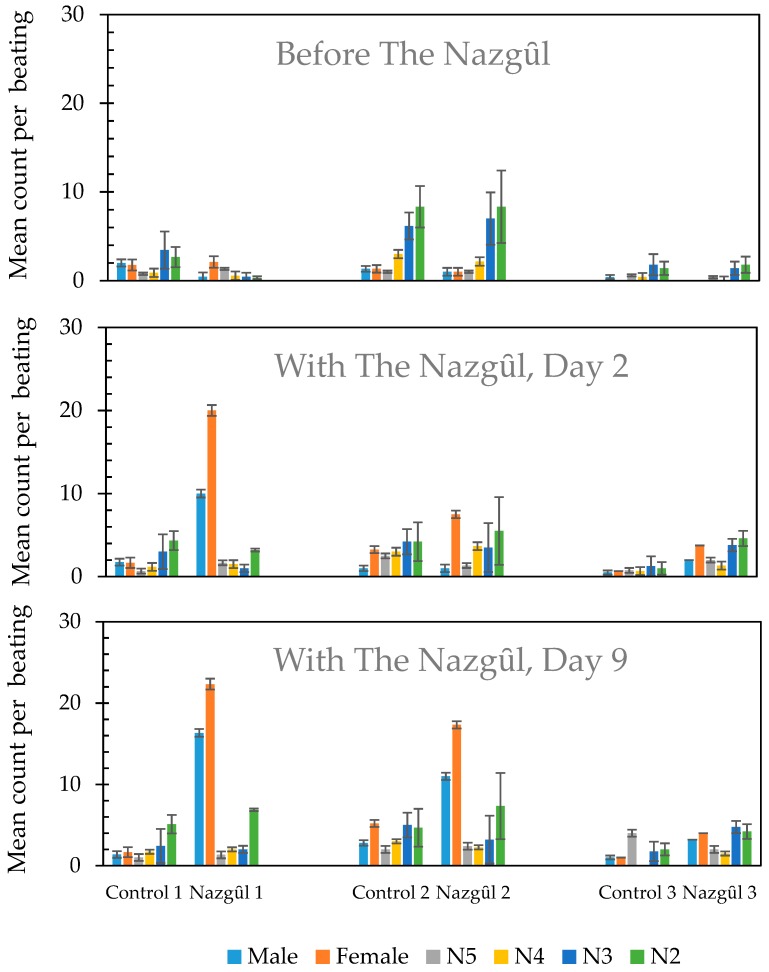
Mean counts of *Halyomorpha halys* by mobile life stage and treatment after beating foliage for two-people-minutes in the three hedgerow transects in apple orchards near Rovereto, Italy. Upper: before the establishment of The Nazgȗl traps, and Lower: Days 2 and 9 after The Nazgȗl were placed at 10 m spacings at three sites (1, n = 9 Nazgȗl; 2, n = 6 Nazgȗl; and 3, n = 5 Nazgȗl). Nazgȗl consist of long-lasting insecticide netting (1.5 × 2 m) and high-loading aggregation pheromone lures, attractive to adults and nymphs (N2–N5). Error bars show standard errors.

**Figure 4 insects-10-00433-f004:**
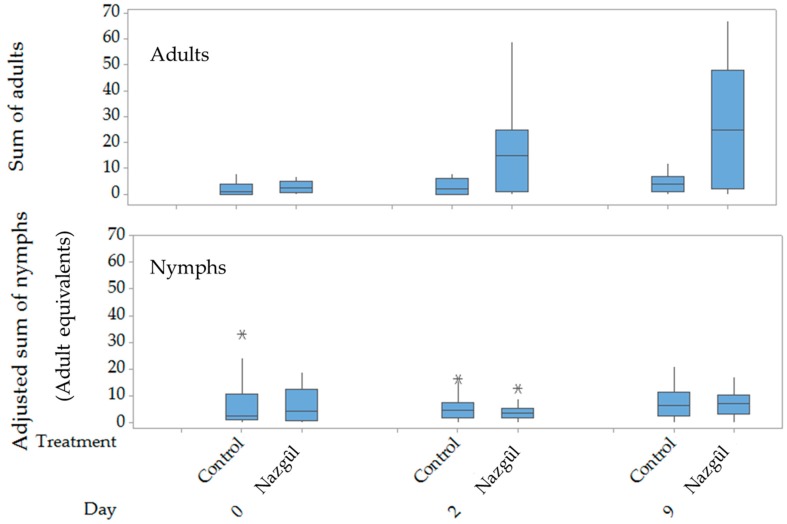
Boxplots of counts of adult (**top**) and adjusted counts of nymphs (**bottom**) *Halyomorpha halys*, sampled by beating foliage for two-people-minutes. Nymph counts were adjusted for stage-specific survivorship to adult [41], and instar counts summed to yield adult equivalents, before and 2 or 9 days after the establishment of Nazgȗl traps at 10 m spacings at three apple orchards near Rovereto, Italy. The asterisks are outliers at least 1.5 times the interquartile range (Q3–Q1).

**Figure 5 insects-10-00433-f005:**
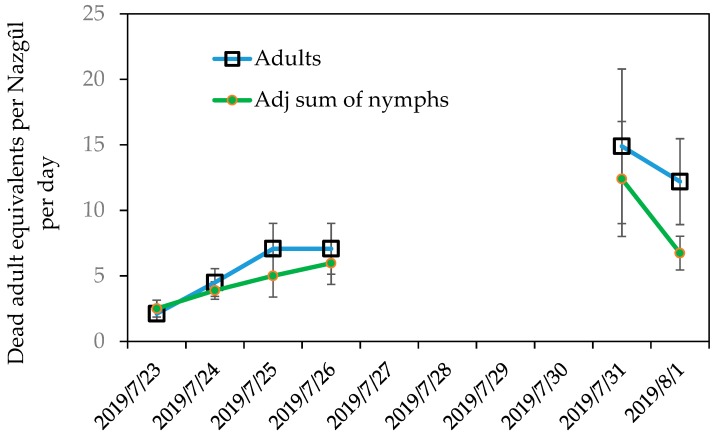
Mean daily counts of dead adults and adjusted sum of dead nymphs after raw counts per instar were converted to adult equivalent *Halyomorpha halys* collected in Nazgȗl trays (n = 20, y = 1.9549 × −85362, r² = 0.7555). Nymph counts were adjusted for stage-specific survivorship to adult [41] and summed across pre-adult stages to yield total adult equivalents. Error bars show standard errors.

**Figure 6 insects-10-00433-f006:**
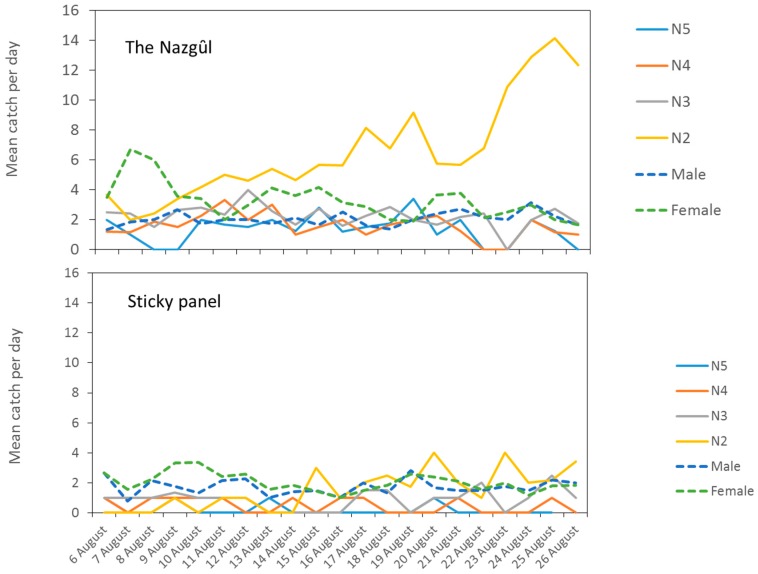
A three-week snapshot of daily phenology of *Halyomorpha halys* recorded using two trapping methods, The Nazgȗl trap (**upper**) or sticky panels (**lower**), baited with aggregation pheromone at a vineyard–forest margin in San Michele all’Adige (TN), Italy (5–26 August 2019).

**Figure 7 insects-10-00433-f007:**
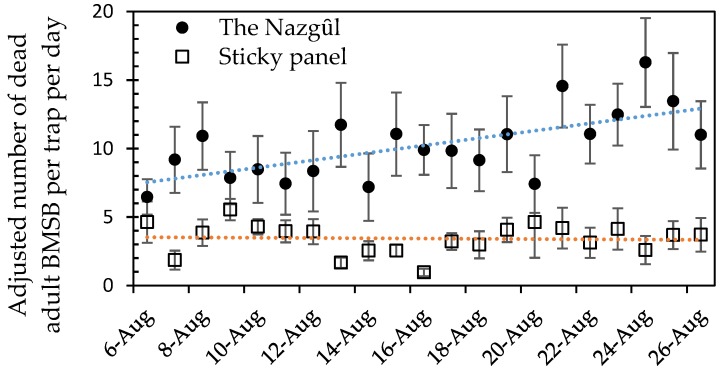
Mean daily count of adjusted adult *Halyomorpha halys* in sticky panels or Nazgȗl traps (Nazgȗl count y = 0.2691 × −11746, R² = 0.4282, *p* < 0.001). Nymph counts were adjusted for stage-specific survivorship to adult [41] and summed with adults to yield total adult equivalents. Error bars show one standard error (*n* = 9 replicates).

**Figure 8 insects-10-00433-f008:**
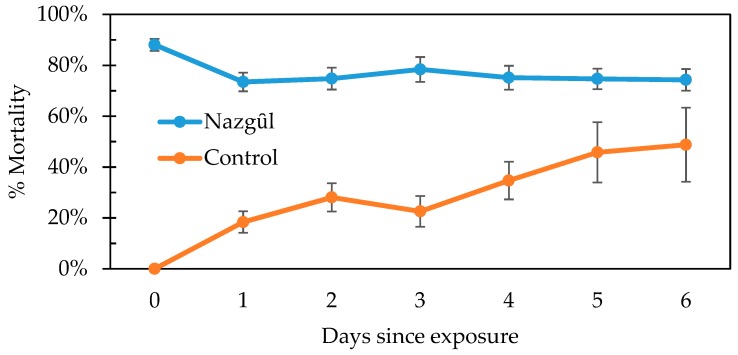
Change in mortality at 24 h due to recovery from pyrethroid knock-down of field-collected *Halyomorpha halys* removed from The Nazgȗl and caged with food and water at 24–26 °C. Controls were field-collected adults. Error bars show one standard error (n = 17 replicates).

**Table 1 insects-10-00433-t001:** Mean catch per trap per day by trap type of all life stages of *Halyomorpha halys* at the forest–vineyard margin in San Michele All’Adige TN, from 10 August 2018–24 September 2018. Two trap types were deployed, Rescue and sticky panel, and two pheromone loadings, standard and high (4 × standard loading).

Lure and Trap	Catch per Trap per Day	SEM
Standard rate/Rescue	1.55	0.69
High rate/Rescue	4.29	1.96
Standard rate/sticky panel	0.62	0.39
High rate/sticky panel	2.67	0.81

**Table 2 insects-10-00433-t002:** The results from a general linear model GLM with a mixed model ANOVA: Trap × Pheromone (Ph) (as fixed effects) × Time (as a random effect) (means in Table 1). Significant values are indicated with an asterisk.

Parameter	df	MS	F	*p*
Trap	1	1.33	8.16	0.068
Ph	1	6.96	42.7	0.003 **
Time	3	0.21	1.28	0.516
Trap × Ph	1	0.17	1.07	0.043 *
Time × Ph	3	0.09	0.52	0.096
Trap × Ph × Time	3	0.15	0.07	0.974

Df: degrees of freedom, MS: mean squares, F: F-ratio, p: probability

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
