# Peer review of "Trapping Brown Marmorated Stink Bugs: “The Nazgȗl” Lure and Kill Nets"

_insects, 2019, doi:10.3390/insects10120433_

Round 1

Reviewer 1 Report

Trapping Brown Marmorated Stink Bugs: Lure and kill using “The Nazgul”.

The work is interesting and takes into consideration a very important and harmful insect to agriculture and human activities.

The field surveys have been well organized even if the activities carried out in a restricted temporal space do not allow to have an overall view of the problem.The results obtained allowed the authors to make some deductions that do not support the hypothesis of practical application with the lure and kill strategy capable of controlling or eradicating an infestation of the Halyomorpha, whichis usually abundant and simultaneously present on many plants. This situation is particularly evident in the final part of the Discussion chapter where the authors put forward many hypotheses and raise many doubts that cannot be answered by the preliminary results of the present work. My suggestion is to accept the manuscript by making profound changes to the text and by limiting the comments to the results of the field trials. Some specific comments are listed below.Lines 2-3: please add preliminary resultsLine 74: add referenceLines 72-78: be more clearLine 100: explain why you decided 1.8 cm and not another heightLine 113: alpha-cypermethrin could be have a repellent effectLine 116: add )Line 120: add ItalyLine 131: 22 July 2019 ?Line 137: can insects trapped produce an allarm signal?Line 145: 5-26 August 2019 ?Figure 1 is not clear, for readers it is impossible to understand how the trap is made. Add information on insecticide persistence and loss of efficacy in case of rain.Line 183: Results and CommentsLine 188: All’AdigeLine 212: … and if you have 8 captures?Line 247: life stage, it is better to use life ageLines 258-263: not clearLine 265: Fig. 6 must be Figure 6Fig. 6  change the charactersFigure 8 change the charactersLines 305-322: this part can be reducedLines 361-362. ???Lines 378-389: not clear to meLines 390-425: this part must be rewritten because it is too vague

Author Response

Rev 1

The work is interesting and takes into consideration a very important and harmful insect to agriculture and human activities. The field surveys have been well organized even if the activities carried out in a restricted temporal space do not allow to have an overall view of the problem. The results obtained allowed the authors to make some deductions that do not support the hypothesis of practical application with the lure and kill strategy capable of controlling or eradicating an infestation of the Halyomorpha, which is usually abundant and simultaneously present on many plants. This situation is particularly evident in the final part of the Discussion chapter where the authors put forward many hypotheses and raise many doubts that cannot be answered by the preliminary results of the present work. My suggestion is to accept the manuscript by making profound changes to the text and by limiting the comments to the results of the field trials. 

We are highly grateful for these comments and have made a major revision.

Some specific comments are listed below.

Lines 2-3: please add preliminary results

Not clear but there is an error in line number

Line 74: add reference

Word added (below).

Lines 72-78: be more clear

Line 100: explain why you decided 1.8 cm and not another height

Convenient and effective from experience.

Line 113: alpha-cypermethrin could be have a repellent effect

Repellency requires contact as pyrethroids like cypermethrin are not volatile. The repellency occurs after contact (Suckling 1983). It shows no sign of this in BMSB since they do sit and die.

Line 116: add )

Done

Line 120: add Italy

Done

Line 131: 22 July 2019 ?

Correct

Line 137: can insects trapped produce an allarm signal?

Yes but would not have a durable effect.

Line 145: 5-26 August 2019 ?

Yes

Figure 1 is not clear, for readers it is impossible to understand how the trap is made.

The words have been revised – it is black cloth on a coat hanger with a catch box. The video has better views of the trap than the photo.

Add information on insecticide persistence and loss of efficacy in case of rain.

Added: The manufacturer’s claimed outdoor insecticidal longevity of the netting is two years and our trials were all short duration.

The netting and lures were draped over and tied to inexpensive wire coat hangers (Nano non-slip hanger, Eurobazar, SRL, Arco, Italy), in order to increase the surface area exposed for landing and walking from nearby foliage.

Line 183: Results and Comments

Revised

Line 188: All’Adige

Line 212: … and if you have 8 captures?

Corrected to ­< 8

Line 247: life stage, it is better to use life age

Unclear what is meant by reviewer, now reads: attractive to all mobile life stages including nymphs (Nx).

Lines 258-263: not clear

The increases in adjusted adult bug counts were 4.3-fold higher than in the untreated plots after two days, and still 3.1-fold higher after nine days of the Nazgȗl present.

Line 265: Fig. 6 must be Figure 6 Fig. 6  change the characters

            OK

Figure 8 change the characters

OK

Lines 305-322: this part can be reduced

Lines 361-362. ???

Paragraph revised

Lines 378-389: not clear to me

Revised

A much larger scale trial of longer duration would be needed for this, similar to trials in USA [21]

Lines 390-425: this part must be rewritten because it is too vague

This has been rewritten with more detail and specifics.

Reviewer 2 Report

The authors present a combination of the aggregation pheromone with an insecticide treated net for monitoring of invasive Halyomorpha halys in Italy. The small study is sound, methods and statistical analysis are appropriate. The conclusion that the presented system could be used in eradication attempts of this invasive bug is far from the results. This sounds very optimistic for me as the numbers of bugs in the surroundings of the traps are increasing. Thus, this aspect should be deleted or better explained.

Minor comments:

Title: The title is not clear as nobody may understand what "The Nazgul" means in this context. I propose to extend the tile like this: Monitoring  Brown Marmorated Stink Bug in Italy using a lure and kill system called “The Nazgȗl”.

Lines 68-70 Citation is missing.

Lines 108-110 Method description remains unclear how the standard protocol for beating tray sampling was performed: How many beats per sample, how many twigs/trees were sampled, beating with a pole? Why the catches have been pooled? This causes a loss of important information like standard deviation etc.

Lines 509-512 The citations no 23 and 24 are incomplete. Please correct:

Peterson, H.; Ali, J.; Krawczyk, G. Biocontrol of the invasive brown marmorated stink bug Halyomorpha halys. In: Pheromones and Other Semiochemicals in Integrated Production and Integrated Protection of Fruit Crops. IOBC-WPRS Bulletin 2019, 146, 119-120. Krawczyk, G.; Morin, H.; Hirst, C. Alternative methods to manage brown marmorated stink bug, Halyomorpha halys. In: Pheromones and Other Semiochemicals in Integrated Production and Integrated Protection of Fruit Crops. IOBC-WPRS Bulletin 2019, 146, 114-118.

Author Response

Rev 2

The authors present a combination of the aggregation pheromone with an insecticide treated net for monitoring of invasive Halyomorpha halys in Italy. The small study is sound, methods and statistical analysis are appropriate. The conclusion that the presented system could be used in eradication attempts of this invasive bug is far from the results. This sounds very optimistic for me as the numbers of bugs in the surroundings of the traps are increasing. Thus, this aspect should be deleted or better explained.

I agree our proposal is far from part of the eradication plan anywhere. I will go for better explaining. The number of people working on arthropod eradications is low. Eradication is cheaper if it can be effected reliably.

The context of places without the bug is different and it appears hard for people in continental USA to relate to eradication, but I have cited some reviews like the Ann Rev Ent by Liebhold et al. I have also referred to the need to delimit the population before eradication – in the following places:

Reports from USA suggest that long-lasting insecticide nets have generally not reduced fruit damage (G. Krawczyk pers. comm.), but improved deployment systems with high treatment density might warrant investigation for delimited populations.

Conclusions

Such killing stations, as demonstrated by the Nazgȗl, could prove useful in suppression or eradication of a delimited population, with or without the trays to determine the numbers killed.

… it can be concluded that this lure and kill concept could be compatible with the needs of an eradication to reduce numbers of a known and delimited population, while avoiding broadcast insecticide use.

Minor comments:

Title: The title is not clear as nobody may understand what "The Nazgul" means in this context.

I propose to extend the tile like this: Monitoring  Brown Marmorated Stink Bug in Italy using a lure and kill system called “The Nazgȗl”.

New title: “Trapping brown marmorated stink bugs: “The Nazgȗl” as lure and kill nets”

Lines 68-70 Citation is missing.

            added

Lines 108-110 Method description remains unclear how the standard protocol for beating tray sampling was performed: How many beats per sample, how many twigs/trees were sampled, beating with a pole? Why the catches have been pooled? This causes a loss of important information like standard deviation etc.

It was beating for time (1 m in 5 m linear hedgerow), not hit number. We had 4 people to do 40 sites or 80 mins beating samples a day (and other things). This is summarised as:

Catches were pooled from the two beating trays to give a single value per plot because of some variation in team personnel sampling efficiency which we did not quantify.

Lines 509-512 The citations no 23 and 24 are incomplete.

Done.

Please correct:

Peterson, H.; Ali, J.; Krawczyk, G. Biocontrol of the invasive brown marmorated stink bug Halyomorpha halys. In: Pheromones and Other Semiochemicals in Integrated Production and Integrated Protection of Fruit Crops. IOBC-WPRS Bulletin 2019, 146, 119-120.

Krawczyk, G.; Morin, H.; Hirst, C. Alternative methods to manage brown marmorated stink bug, Halyomorpha halys. In:

OK

Reviewer 3 Report

The authors ostensibly represent their study as development of “lure and kill” for brown marmorated stink bug. Though the authors conduct four experiments, each experiment is short with few replicates, and the combined total sampling time for all the experiments do not exceed 2 months and 1 week. Trapping field studies less than two and certainly less than one year do not rise to the level of standalone publications. The authors demonstrate their lack of awareness of existing literature on brown marmorated stink bug, especially in reference to the only other published studies on a lure and kill system for the brown marmorated stink bug in the USA, but also show their ignorance in other areas. The authors overstate their data, make extrapolations beyond the very limited data that they present, and come to conclusions that are not supported by the data they do show the reader.  Finally, the authors use a strawman argument by comparing their “lure and kill” system to sticky traps with high dose lures; however, sticky cards with lures was never meant to be a “lure and kill” system, but rather a monitoring and surveillance system. The difference is in whether one is used for monitoring or management, and the sticky cards were never intended to be used beyond monitoring. Thus, the central argument of the paper again demonstrates the authors lack of awareness of intent of tools for BMSB. I believe the approach taken by the authors has merit, but requires more data and must demonstrate a better awareness of prior work on BMSB.  I have the following specific comments:

Line 47-48 throughout the paper, the authors make a point of saying their new tools are for “eradication”. However, eradication of invasive species have been exceedingly rare, and is usually only effective when populations are so low that they are almost below detectable levels. This is especially a weird emphasis for an invasive species that has so perniciously invaded new areas and has NEVER been eradicated from any location. It just takes one surviving female for the population to continue, and no tool is 100% effective.

Line 49-50 throughout the MS, all S and R notations need to be italicized.

Line 56 here and throughout, the established terminology for this technology is long-lasting insecticide-incorporated netting (NOT long “life” netting). See for example the first published study on LLIN in the literature for BMSB: Kuhar et al. 2017 J Econ Entomol 110: 543-545 (https://academic.oup.com/jee/article/110/2/543/3060404).

Line 86 the authors cite 20 mg of pheromone and 200 mg of MDT as their “lure and kill” dosages. These are not “lure and kill” relevant dosages. The only prior study on attract-and-kill used a dose of ~840 mg of aggregation pheromone, so the authors are only 42-fold off. The authors are working with a high monitoring dose of pheromone.

Seems odd that the authors do not mention the only prior documented lure and kill approach for BMSB. Morrison et al. 2016 J Pest Science 89: 81-96 (https://idp.springer.com/authorize/casa?redirect_uri=https://link.springer.com/article/10.1007/s10340-015-0679-6&casa_token=MYiQxQnfvCkAAAAA:FOoL_6uc74Q-Hw-Ob4AJPJk-02m22jJ-EaTfM8a59uOpU_amjWZ82CyVmkr5HKdgFmGL9ovHj_OOTt2c), and Morrison et al. 2019 Successful management of Halyomorpha halys (Hemiptera: Pentatomidae) in commercial apple orchards with an attract‐and‐kill strategy

 Pest Manage. Sci. 75: 104-114 (https://onlinelibrary.wiley.com/doi/full/10.1002/ps.5156).

Line 95 the authors’ first experiment only has three samples—unacceptable. Seems like none of these experiments really fit together and have been cobbled together for another publication.

Line 116 insert missing parenthesis

Line 129 authors indicated that they released adults after counting? Seems like this would inflate their estimates of captures and how effective their traps are.

Line 135-6 trays were emptied—presumably right in the area near the trap so they could be recaptured?

Line 142 throughout the MS, make the “s” in “Sticky” lower case—it is not a proper noun.

Line 150-51 how far away were controls? Were they really representative of the area that the traps were located given how patchy BMSB nymphs and adults can be in the landscape?

Line 158 the authors seem to have arbitrarily delimited their data into three different capture categories—but why? And, how was 8 considered the magical cutoff point? The whole correspondence analysis seems excessive and unnecessary for the primary focus of this study.

For the stats analysis—why are the authors transforming data (old-fashioned) when they should be using a generalized linear model with appropriate distribution that fits the data without torturing it to fit assumptions?

Lines 168-169 The authors present their data in a very misleading fashion by somehow magically converting nymphal captures into “adult equivalents” based on uncertain assumptions that may differ for their specific study area, which is not typically done in BMSB research (or any trapping research). It would be much more useful (and less misleading) if the authors present the nymphal data in addition to those for the adults. My hunch is that adult captures were low, and the authors wanted to bolster the case for their so-called “lure and kill” system, so tried to inflate the numbers. This is not okay.

Table 2: the degrees of freedom for the denominator need to be added; it concerns me that they were not—perhaps because of the low number of replicates.

Figure 2: not sure this adds much to the MS.

Figure 3: again, not sure the beat samples are relatable to the traps, unless the authors specify where they were taken.

Figure 4: There are no measures of variation on any of the graphs. Standard errors need to be added to every bar.

Line 246-7 no way to verify this statement because the authors do not present data for the nymphs.

Line 239-40 capturing ~600 adults (but not all adults, because presumably some are “adult equivalents”) during the peak BMSB season is not that impressive, esp given the number of samples. Large pyramid traps will typically capture ~150-200 adults per week during peak BMSB season. The Nazgul is a monitoring trap, not a “lure and kill” trap

Figure 5 & 6: need to add standard errors.

From Figure 6, it looks like the Nazgul are killing 10 adults per day. I wonder if this rate would continue over a week, though—clearly the authors imply that it does through their extrapolated line, but this would need to be confirmed, and cannot given the authors small dataset.

Line 297 20% recovery is significant, but the authors try to downplay it. It would be more helpful if the authors discuss how this limitation could be overcome, either by improving the netting, or by deployment of trap.

Line 298 how do the authors know this? Did they make standardized observations of traps? An example of extrapolating beyond the data.

Table 3: it would be more helpful to the reader to see the actual mortality over time. Notably, the authors left out recovery after 48 and 72 h, even though they specify collecting it on Line 149.

Line 309 again with the fixation on eradication.

Line 331 authors say effectiveness would be increased under cooler temperatures, but would it really? Stink bugs do not fly under 20C, and insect movement declines exponentially with decreasing temperature, so likely there would be a lower encounter rate. The authors need to moderate their language around this issue—especially since they don’t present diurnal data nor relate it meaningfully to local temperatures at field sites.

Line 335 authors discuss effect of plume reach, but do not cite published studies on plume reach, including Kirkpatrick et al. 2019 48: 1104-1112 (https://academic.oup.com/ee/article/48/5/1104/5554151).

Line 351-2 again, the authors do not meaningfully relate back their work to the prior literature on BMSB, even when mentioning items that have been directly researched. Authors discuss host volatiles, but applicable articles on this that were left out are Blaauw et al. 2019 Plant Stimuli and their Impact on Brown Marmorated Stink Bug Dispersal and Host Selection Frontiers in Ecology and Evolution (https://www.frontiersin.org/articles/10.3389/fevo.2019.00414/abstract) and Morrison et al. 2017 Agric For Entomol 20: 62-72 (https://onlinelibrary.wiley.com/doi/full/10.1111/afe.12229?casa_token=t1yI5muSZSMAAAAA%3AyA9H0p3fJpjEHl1A2y7AUNnstZVAVpD5S0mfCvbxakZ9Yw5pIpVL79nqcWy3Tqe-bbI4Uc7CKVkjLg).

Line 355 Catching 10 adults/day for four days does not qualify the Nazgul as a “lure and kill” trap. It seems like more data is needed.

Line 361 impossible to verify results with nymphs since the data are not presented.

Line 368 this seems like preliminary work, and the authors confirm it here.

Line 370 where is the data supporting this statement?

Line 375-6 citation or data?

Line 385 everywhere else in the MS, the authors use the term knockdown, but here use a different word. It needs to be defined or deleted.

Line 407-8 the authors state that the sticky cards are not suitable for tracking phenology, but this is patently untrue according to prior published literature. A study in 18 US States and 115 sites demonstrated their usefulness in tracking all life stages of BMSB for reliable monitoring: Acebes-Doria et al. 2019 J Econ Entomol (https://academic.oup.com/jee/advance-article/doi/10.1093/jee/toz240/5564854).

Author Response

Dear Editor

Thank you for the opportunity to take on the reviewers comments with a major revision, which we agree was warranted. Most points have been addressed, although Reviewer 3 had a problem with the duration of the study which could not be addressed with new data since it involves sabbatical leave over two summers in Italy. Instead of this view that season-long studies are the only option to progress this technology, the compressed studies here were designed to address a lot in a short time and this context is now better explained.

This urgency is necessary because globalization is spreading this bug and it threatens new jurisdictions that are not only aware of it from work in USA and elsewhere, but counties like New Zealand want solutions urgently. We have expanded the discussion of the context of eradication in response to the third reviewer, which relates to the many countries that do not yet have BMSB. We cited one paper recommended by Reviewer three but we noted that only the abstract is available on line at this time in Frontiers. We have also provided citations of two new BMSB papers in this same journal that are related to the concept of eradication of BMSB, on live traps and the Sterile Insect Technique. We have added a new figure on the mortality of BMSB in response to Reviewer three, dropped one figure and made improvements to others including the standard error bars requested. We hope that the changes are acceptable to address the points raised in all the useful feedback from all three reviewers.

This paper also has a video abstract.

Kind regards

DM Suckling

Rev 3

The authors ostensibly represent their study as development of “lure and kill” for brown marmorated stink bug. Though the authors conduct four experiments, each experiment is short with few replicates, and the combined total sampling time for all the experiments do not exceed 2 months and 1 week.

Trapping field studies less than two and certainly less than one year do not rise to the level of standalone publications.

This depends on the purpose and the urgency, which is great for countries without BMSB. Even short trials can provide useful information for decision-makers. Since my last paper with a new BMSB trap I have had the New Zealand government asking about it and the New Zealand apple industry investing in more trapping. There is interest in Chile and other countries in this work.

The authors demonstrate their lack of awareness of existing literature on brown marmorated stink bug, especially in reference to the only other published studies on a lure and kill system for the brown marmorated stink bug in the USA, but also show their ignorance in other areas.

            The reviewers suggested references have been adopted in full.

The authors overstate their data, make extrapolations beyond the very limited data that they present, and come to conclusions that are not supported by the data they do show the reader. 

The paper has been revised.

Finally, the authors use a strawman argument by comparing their “lure and kill” system to sticky traps with high dose lures; however, sticky cards with lures was never meant to be a “lure and kill” system, but rather a monitoring and surveillance system. The difference is in whether one is used for monitoring or management, and the sticky cards were never intended to be used beyond monitoring. Thus, the central argument of the paper again demonstrates the authors lack of awareness of intent of tools for BMSB.

The text has been revised. They are certainly planned for more than monitoring in New Zealand with the additional role of detection and delimitation (implying capacity for detection of absence without error).

Various trapping systems have been investigated for IPM [24], and based on these lures, surveillance systems have been tested in the context of border protection [9]. The lures have also been tested for lure and kill at high doses (84 and 840 mg) in sacrificial insecticide-treated trees [25]. Tree-level attract and kill has potential to reduce damage in orchards but so far is very expensive [26]. We chose to use sticky panels as a reference system due to extensive previous testing [27], for comparison with alternative traps under an expanding BMSB population in the north of Italy. While there is no proposal to use sticky traps as a control tool using mass trapping, our goal was to determine what improvements could be made to support an incursion response. Novel systems are especially needed for border protection in countries with high interception rates such as New Zealand [9], where interceptions of alien pests are often detected in urban or peri-urban areas.

In discussion: Of course, sticky panels are not normally considered for suppression.

I believe the approach taken by the authors has merit, but requires more data and must demonstrate a better awareness of prior work on BMSB.  

New data on mortality was added. Account was taken of the papers recommended. An error in stating that three trap readings were made was corrected.

New text: Finally, in Experiment 4 the kill efficacy of Nazgȗl was compared with sticky panels, using the same lures (normally used for monitoring), and post-insecticide exposure recovery was investigated.

I have the following specific comments:

Line 47-48 throughout the paper, the authors make a point of saying their new tools are for “eradication”. However, eradication of invasive species have been exceedingly rare, and is usually only effective when populations are so low that they are almost below detectable levels.

The reviewer is referred to a global eradication database listing New Zealand as having conducted 92 arthropod or plant pathogen eradications, with 337 in USA (www.b3nz.org/gerda). Not that rare ?

This is especially a weird emphasis for an invasive species that has so perniciously invaded new areas and has NEVER been eradicated from any location. It just takes one surviving female for the population to continue, and no tool is 100% effective.

The text has been revised to elaborate on eradication and the correct context for this work, which is unfamiliar to the reviewer. IF Allee effects exist in this species, it could be possible to have extinction if the density is too low (but above 1).

Adult BMSB is estimated to have the capacity to fly 2-3 km per day [10] and generally has a high natural dispersal ability [11-13], which is likely human-assisted considering how fast it spread across Europe and North America. Governments considering eradication of arthropods are facing increasing challenges from novel pest biodiversity [14]. Avoiding the long term problem of BMSB by eradication warrants investigation where this might be possible [15], but this approach requires the development of suitable tools [16]. No eradication attempts have been reported thus far but New Zealand has taken the innovative and proactive step of approving an exotic biological control agent (Trissolcus japonicus) for release in the event of BMSB establishment [17]. Host range testing suggests significant risks to several non-target pentatomid bugs from release of this egg parasitoid [17]. New Zealand has an official government goal of harnessing 4.7 million pairs of eyes for biosecurity [18], and a long track record of unwanted arthropod eradication attempts [19], as well as a desire to avoid non-target impacts from new parasitoids if possible [20], leading to our search for possible alternative eradication technologies, such as the sterile insect technique [21].

The reviewer is also referred to Vandervoet et al. 2019 on surveillance for first detection of BMSB. The text has been expanded and the concept of delimited populations has been emphasised (see above). The authors agree that the jury is out whether the Nazgȗl could make a difference (see video) but this paper is putting the technology in that context for the first time. It is currently not on the agenda. It should be part of the response planning and this includes pre-registration of the technology (no similar products will mean delays to access).

Line 49-50 throughout the MS, all S and R notations need to be italicized.

Done

Line 56 here and throughout, the established terminology for this technology is long-lasting insecticide-incorporated netting (NOT long “life” netting). See for example the first published study on LLIN in the literature for BMSB: Kuhar et al. 2017 J Econ Entomol 110: 543-545 (https://academic.oup.com/jee/article/110/2/543/3060404).

Now cited

Line 86 the authors cite 20 mg of pheromone and 200 mg of MDT as their “lure and kill” dosages. These are not “lure and kill” relevant dosages. The only prior study on attract-and-kill used a dose of ~840 mg of aggregation pheromone, so the authors are only 42-fold off. The authors are working with a high monitoring dose of pheromone.

This has been noted.

Seems odd that the authors do not mention the only prior documented lure and kill approach for BMSB. Morrison et al. 2016 J Pest Science 89: 81-96 (https://idp.springer.com/authorize/casa?redirect_uri=https://link.springer.com/article/10.1007/s10340-015-0679-6&casa_token=MYiQxQnfvCkAAAAA:FOoL_6uc74Q-Hw-Ob4AJPJk-02m22jJ-EaTfM8a59uOpU_amjWZ82CyVmkr5HKdgFmGL9ovHj_OOTt2c), and Morrison et al. 2019 Successful management of Halyomorpha halys (Hemiptera: Pentatomidae) in commercial apple orchards with an attract‐and‐kill strategy

 Pest Manage. Sci. 75: 104-114 (https://onlinelibrary.wiley.com/doi/full/10.1002/ps.5156).

           This has also been cited.

Line 95 the authors’ first experiment only has three samples—unacceptable.

There was an error, it was four readings. This comment ignores the five replications – so there are 20 readings not three.

Seems like none of these experiments really fit together and have been cobbled together for another publication.

The other reviewers didn’t say that and it is hard to see why this reviewer thinks that, since the studies follow a progression.

Line 116 insert missing parenthesis

Done

Line 129 authors indicated that they released adults after counting? Seems like this would inflate their estimates of captures and how effective their traps are.

No, this only keeps the subsequent catch distributed between trap types, once they  were installed, but allows us to have an estimate of this density.

Line 135-6 trays were emptied—presumably right in the area near the trap so they could be recaptured?

Yes, revised

Beating tray samples were conducted as per Experiment 2, with the BMSB counted and adults identified to sex and nymphs to life stage before being released again.

Line 142 throughout the MS, make the “s” in “Sticky” lower case—it is not a proper noun.

Done

Line 150-51 how far away were controls? Were they really representative of the area that the traps were located given how patchy BMSB nymphs and adults can be in the landscape?

           Maximum 80m. The controls comprised BMSB hosts.

Line 158 the authors seem to have arbitrarily delimited their data into three different capture categories—but why? And, how was 8 considered the magical cutoff point? The whole correspondence analysis seems excessive and unnecessary for the primary focus of this study.

         It was the median. The figure has been removed.

For the stats analysis—why are the authors transforming data (old-fashioned) when they should be using a generalized linear model with appropriate distribution that fits the data without torturing it to fit assumptions?

     The transformation was very effective at describing the data so a GLM was not necessary.

Lines 168-169 The authors present their data in a very misleading fashion by somehow magically converting nymphal captures into “adult equivalents” based on uncertain assumptions that may differ for their specific study area, which is not typically done in BMSB research (or any trapping research).

It would be much more useful (and less misleading) if the authors present the nymphal data in addition to those for the adults. My hunch is that adult captures were low, and the authors wanted to bolster the case for their so-called “lure and kill” system, so tried to inflate the numbers. This is not okay.

This has been done for Experiments 3 and 4 where nymphs were studied.

The data are presented both ways, nymphs and adults separate or summed after correction. Adult captures were not low for the area. This approach is innovative but is science-based. Killing nymphs is actually significant since they can’t then be adult pests. Their probability at survival to do that is simply taken into account. This is okay.

Table 2: the degrees of freedom for the denominator need to be added; it concerns me that they were not—perhaps because of the low number of replicates.
Added to Methods for Expt 1: The denominator degrees of freedom (calculated by subtracting the number of sample groups from the total number of samples tested) is 16.

Figure 2: not sure this adds much to the MS.

Removed

Figure 3: again, not sure the beat samples are relatable to the traps, unless the authors specify where they were taken.

Done … were operated continuously for 1 minute per plot (0.5-5m either side of the lure)

Figure 4: There are no measures of variation on any of the graphs. Standard errors need to be added to every bar.

This has been done

Line 246-7 no way to verify this statement because the authors do not present data for the nymphs.

See new Fig 3

Line 239-40 capturing ~600 adults (but not all adults, because presumably some are “adult equivalents”) during the peak BMSB season is not that impressive, esp given the number of samples. Large pyramid traps will typically capture ~150-200 adults per week during peak BMSB season.

The density is highly likely to be different (presumably higher) at the expert reviewers location, where presumably the establishment history is longer than in the Trentino.

The Nazgul is a monitoring trap, not a “lure and kill” trap

It is discussed as doing both functions in the paper. Future use patterns are hard to predict. It is a lot of work as a monitoring trap !

Figure 5 & 6: need to add standard errors.

This has been done. Error bars show standard errors.

From Figure 6, it looks like the Nazgȗl are killing 10 adults per day. I wonder if this rate would continue over a week, though—clearly the authors imply that it does through their extrapolated line, but this would need to be confirmed, and cannot given the authors small dataset.

This point is addressed in the next experiment which lasted 21 days with continuous daily kill. It rose steadily (unlike the sticky panel traps).

Line 297 20% recovery is significant, but the authors try to downplay it. It would be more helpful if the authors discuss how this limitation could be overcome, either by improving the netting, or by deployment of trap.

Acknowledged.

It appears that adult BMSB were removed from the Nazgȗl before lethal exposure in up to 25% of cases, but the duration of exposure would vary with individual arrival time at the trap. Improvements to the netting, such making it’s micro scale texture hairy might help.

Line 298 how do the authors know this? Did they make standardized observations of traps? An example of extrapolating beyond the data.

The sentence has been removed.

Table 3: it would be more helpful to the reader to see the actual mortality over time. Notably, the authors left out recovery after 48 and 72 h, even though they specify collecting it on Line 149.

Replaced with figure.

Line 309 again with the fixation on eradication.

This “fixation” represents a possible use pattern for the technology.

Line 331 authors say effectiveness would be increased under cooler temperatures, but would it really? Stink bugs do not fly under 20C,

No but they can sit on the netting at 5˚C overnight ?

and insect movement declines exponentially with decreasing temperature, so likely there would be a lower encounter rate.

At night ?

The authors need to moderate their language around this issue—especially since they don’t present diurnal data nor relate it meaningfully to local temperatures at field sites.

If they are sitting on the netting overnight the temperature overnight will be important.    

Line 335 authors discuss effect of plume reach, but do not cite published studies on plume reach, including Kirkpatrick et al. 2019 48: 1104-1112 (https://academic.oup.com/ee/article/48/5/1104/5554151).

         Acknowledged thanks.

Line 351-2 again, the authors do not meaningfully relate back their work to the prior literature on BMSB, even when mentioning items that have been directly researched. Authors discuss host volatiles, but applicable articles on this that were left out are Blaauw et al. 2019 Plant Stimuli and their Impact on Brown Marmorated Stink Bug Dispersal and Host Selection Frontiers in Ecology and Evolution (https://www.frontiersin.org/articles/10.3389/fevo.2019.00414/abstract)

Now cited

A forthcoming study has expanded knowledge of the impact of fruit volatiles on dispersal and host location [50], which clearly play a role [51].

and Morrison et al.

2017 Agric For Entomol 20: 62-72 (https://onlinelibrary.wiley.com/doi/full/10.1111/afe.12229?casa_token=t1yI5muSZSMAAAAA%3AyA9H0p3fJpjEHl1A2y7AUNnstZVAVpD5S0mfCvbxakZ9Yw5pIpVL79nqcWy3Tqe-bbI4Uc7CKVkjLg).

Now cited

Line 355 Catching 10 adults/day for four days does not qualify the Nazgul as a “lure and kill” trap. It seems like more data is needed.

There is more data. It lures and it kills (at least somewhat but we don’t fully know) and its value may depend on the local density in an incipient outbreak ? Population suppression is the test of efficacy but large open populations are not the target for this work. We did SIT with 5000 male moths a week on a small tussock moth population. Eradication is a different topic from pest management.

Line 361 impossible to verify results with nymphs since the data are not presented.

Now presented

Line 368 this seems like preliminary work, and the authors confirm it here. Line 370 where is the data supporting this statement?

In Experiment 3, the desired result of a reduction in numbers of BMSB was not seen in the three treated hedgerows, in fact there was a significant increase in BMSB sampled by beating trays in the treated areas (Figures 3 and 4), despite a daily removal of adult and nymphal BMSB (Figure 5).

Line 375-6 citation or data?

Removed

Line 385 everywhere else in the MS, the authors use the term knockdown, but here use a different word. It needs to be defined or deleted.

Done

Line 407-8 the authors state that the sticky cards are not suitable for tracking phenology, but this is patently untrue according to prior published literature. A study in 18 US States and 115 sites demonstrated their usefulness in tracking all life stages of BMSB for reliable monitoring: Acebes-Doria et al. 2019 J Econ Entomol (https://academic.oup.com/jee/advance-article/doi/10.1093/jee/toz240/5564854).

The point is the resolution is low so more sampling effort is needed. People use unbaited sticky panels when they have no choice (no lure for some insects). An improvement of 3.5-fold appears to increase the resolution considerably. This increase of resolution is available if needed.

Round 2

Reviewer 1 Report

The new version of the manuscript has been improved and can be accepted

Reviewer 3 Report

The authors have by and large addressed or seriously considered all my comments. Though the authors did not provide a longer sampling interval, the threat of BMSB is urgent, especially for areas like New Zealand (as mentioned by the authors) and the added context makes it clear why this study is required.  I believe the current contribution is now suitable for publication in Insects.